# Elucidation of the Neuroprotective Effects of Astaxanthin Against Amyloid β Toxicity in the SH-SY5Y Human Neuroblastoma Cell Line

**DOI:** 10.3390/molecules30214271

**Published:** 2025-11-03

**Authors:** Sahithya Hulimane Ananda, Masahiro Kuragano, Kiyotaka Tokuraku

**Affiliations:** Division of Sustainable and Environmental Engineering, Muroran Institute of Technology, Mizumoto 27-1, Muroran 050-8585, Hokkaido, Japan; sahianand0104@gmail.com

**Keywords:** astaxanthin, amyloid β aggregation, Alzheimer’s disease, SH-SY5Y cells

## Abstract

Alzheimer’s disease (AD) is a neurodegenerative disorder characterized by memory loss and cognitive decline, primarily due to amyloid β (Aβ) aggregation in the brain. Astaxanthin (AxN), a xanthophyll carotenoid derived from *Haematococcus pluvialis*, possesses antioxidant and neuroprotective properties. This study investigated the neuroprotective effects of AxN against Aβ aggregation in human neuroblastoma SH-SY5Y cells. Initially, AxN inhibited Aβ aggregation in DMEM/F12 culture medium but not in PBS, suggesting a medium-dependent effect. Using quantum dot nanoprobes, Aβ aggregation was visualized in the presence of SH-SY5Y cells. AxN treatment (0.032–20 µM) significantly reduced Aβ aggregation and accumulation on SH-SY5Y cells. AxN also prevented Aβ-induced early apoptotic cell death but was less effective against late necrosis. Furthermore, a wound-healing assay showed that AxN restored the impaired cell motility caused by Aβ aggregation. Thioflavin T staining confirmed the reduction in Aβ fibril formation around the cells following AxN treatment. In conclusion, our study suggests that AxN prevents Aβ aggregation and accumulation on the cell surface, thereby restoring cell motility and preventing early apoptosis in neuronal cells.

## 1. Introduction

Alzheimer’s disease (AD) is a chronic neurodegenerative disease caused by—as a complex pathogenesis—the abnormal deposition of amyloid β (Aβ) and hyperphosphorylated tau in senile neurons [1,2,3]. AD is primarily characterized by memory loss, cognitive decline, and visual–spatial dysfunction [4,5]. Misfolded Aβ protein forms amyloid plaques, which cause extracellular deposition in between neurons, and intracellular deposition of neurofibrillary tangles (NFTs) in the neurons are formed by tau proteins, leading to neurodegenerative diseases [6,7].

The progressive decline in cognitive functions leads to acute dementia, while oxidative damage induced by reactive oxygen species disrupts cellular processes, leading to impaired protein function while promoting the aggregation of Aβ plaques, a hallmark of AD [8]. Aβ is a neurotoxic polypeptide composed of 39–43 amino acid residues that aggregate into soluble oligomers and insoluble amyloid fibrils [9,10]. Aβ_40_ and Aβ_42_ are the most prevalent species that contribute to Aβ-related neurodegenerative disorders. Aβ_42_ accumulates more rapidly than Aβ_40_, and it is also a primary component of senile plaques, which are characteristic of AD [11,12,13].

Astaxanthin (AxN) is a red-orange pigmented carotenoid that is found in marine organisms and microalgae [14]. It is abundantly produced by the microalgal species, *Haematococcus pluvialis* [15]. AxN is also a potent antioxidant that can prevent neuronal cell death and oxidative stress, and improve spatial memory [7,16]. The potent antioxidant effects of AxN are attributed to the stability of its ionone rings and polyene structure, allowing it to effectively counteract damage by free radicals [17]. It also enhances mitochondrial functions in neurons, reduces DNA damage, and alleviates cellular stress. In recent years, several in vitro and in vivo studies on AxN have demonstrated its neuroprotective effects on brain damage, anti-cancer, anti-inflammatory, anti-diabetic, and immunomodulatory activities [18,19,20,21,22].

Although the exact cause of this disease has not yet been identified, the neuronal damage and death caused by Aβ induce the cognitive deficits that characterize AD [23]. Despite considerable research on AD at present, there is no permanent cure for AD, as the main pathological cause remains unknown. Therefore, in this study, we investigated the mechanism of neuroprotection of AxN against Aβ-induced cytotoxicity using the SH-SY5Y human neuroblastoma cell line.

Following our previous reports [24,25,26], which demonstrated the inhibitory activity of various natural products on Aβ_42_ aggregation, this study aimed to evaluate the effect of AxN on 25 µM Aβ_42_ both in vitro and in a cell line model. To achieve this, we assessed Aβ deposition by visualizing and quantifying aggregates using quantum dots (QDs) and demonstrated that AxN inhibited Aβ aggregation in cells and suppressed early apoptosis. Furthermore, AxN was found to exert a protective effect on Aβ-induced impairments in cell motility and migration dynamics, which are commonly affected in neurodegenerative diseases.

## 2. Results

### 2.1. Inhibitory Effect of Aβ Aggregation by AxN

Initially, we investigated the inhibitory effect of 0.32–200 µM AxN on the aggregation of Aβ using the microliter-scale high-throughput screening (MSHTS) system. After 24 h of incubation in phosphate-buffered saline (PBS), similar aggregates were observed regardless of the concentration of AxN added, whereas in DMEM/F12, the amount of Aβ aggregates was lower in the presence of 200 µM AxN than in the absence of AxN (Figure 1A). We also found that the shape of Aβ aggregates differed between PBS and DMEM/F12, which is presumably due to the various substances found in these media affecting the interaction of Aβ fibrils. The amount of Aβ aggregates was estimated from the standard deviation (SD) values of fluorescent intensities of images. Half-maximal effective concentration (EC_50_) values clearly determined that Aβ deposition was inhibited by AxN in DMEM/F12 but not in PBS as an MSHTS solvent (Figure 1B,C). This result suggests that the inhibitory effect of AxN is enhanced in culture medium but is inefficient in pure buffer conditions.

### 2.2. Attenuation of Aβ Aggregation and Deposition by AxN on SH-SY5Y Neuroblastoma Cells

Similarly, using the QD imaging technique, we investigated the inhibitory effect of Aβ aggregation and deposition on SH-SY5Y neuroblastoma cells in the presence of different concentrations of AxN. Fluorescence images showed the effective reduction in the accumulation of Aβ around the cells in a concentration-dependent manner (Figure 2A). Quantitative analysis was evaluated using ImageJ 1.53e software and showed a significant decrease in fluorescent intensity, indicated by the mean gray value, in the presence of 0.8 µM or more of AxN (Figure 2B). The results show that AxN reduced the deposition of Aβ aggregates around SH-SY5Y cells.

### 2.3. Inhibitory Effects of AxN on Aβ Deposition in SH-SY5Y Neuroblastoma Cells

We performed the Thioflavin T (ThT) assay to study the effect of AxN on the formation and deposition of Aβ fibrils. ThT is a fluorescent dye used to detect and quantify the formation of β-sheet-rich Aβ fibrils. The decrease in the level of fluorescence intensity of ThT at 24 h demonstrated the inhibitory effect of 20 µM AxN on Aβ deposition (Figure 3A). Fluorescence images of ThT were also captured using an inverted fluorescence microscope (Figure 3B). The images were analyzed using ImageJ software, which demonstrated the inhibition of Aβ deposition by different concentrations of AxN (Figure 3C). AxN at 20 µM showed a significant anti-aggregation effect with the greatest reduction in the intensity of ThT fluorescence. These results indicate that AxN can reduce the deposition of Aβ fibrils on the cell surface.

### 2.4. Effects of AxN on Aβ-Induced Apoptosis in SH-SY5Y Neuroblastoma Cells

We analyzed in detail the effect of 25 µM Aβ in the presence or absence of AxN (0.032–20 µM) on apoptosis in human neuroblastoma cells. Cells treated with AxN reduced the number of apoptotic cells in a concentration-dependent manner (Figure 4A, p-SIVA). Quantification of fluorescence intensity demonstrated that the early apoptotic cell death induced by Aβ was significantly suppressed by the addition of 20 µM AxN (Figure 4B). On the other hand, the late necrosis induced by Aβ aggregation was not fully suppressed by AxN (Figure 4C). This result suggests that AxN shows a neuroprotective effect in SH-SY5Y cells by inhibiting apoptosis caused by Aβ aggregation and deposition.

### 2.5. Effects of AxN on SH-SY5Y Neuroblastoma Cell Motility

We recently reported that aggregation around cells affected cell motility [11,27]. To elucidate the effect of Aβ on the motility of SH-SY5Y cells, we performed a wound healing assay (WHA). The SH-SY5Y cells, which were treated with 0.032–20 µM AxN in the presence of 25 µM Aβ, and were monitored for 24 h, migrated towards the wound area (Figure 5B). The difference in area covered by cells over 24 h is shown in Figure 5C. Although no significant difference was found, this result suggests a tendency for AxN to rescue the defects of cell motility caused by Aβ aggregation and deposition.

Using Hoechst staining, the migration dynamics of individual SH-SY5Y cells were assessed (Figure 6A). AxN-treated cells significantly improved cell speed (Figure 6C), and directional persistence was partially restored (Figure 6D). This result suggests that 20 µM AxN can attenuate the inhibitory effects of cell migration and its dynamics induced by Aβ aggregation and deposition.

## 3. Discussion

Astaxanthin, a xanthophyll carotenoid primarily found in algae, salmon, shrimps, and other marine organisms, is recognized as a strong and naturally occurring antioxidant. Its antioxidant activity is superior to that of other carotenoids (fucoxanthin, canthaxanthin, zeaxanthin, β-carotene, and α-tocopherol) [28,29]. In this study, AxN was able to reduce the abnormal deposition of Aβ on human neuroblastoma SH-SY5Y cells. Although inhibitory effects via cytotoxicity assays were not confirmed, the prevention of early apoptosis and the rescue of cell migration ability were shown.

First, in this study, we evaluated the Aβ aggregation inhibitory activity of AxN using the MSHTS system (Figure 1), a previously established system to visualize Aβ aggregation and to estimate inhibitory activity using QD nanoprobes [13,25,30]. Interestingly, when PBS and DMEM/F12 were used as solvents for the MSHTS system, AxN was shown to inhibit Aβ aggregation in DMEM/F12 but not in PBS (Figure 1). It has been reported that DMEM/F12 provides a stable environment that enhances the antioxidant property of AxN, whereas PBS does not contain biological components [31]. Similarly, DMEM/F12 presumably contains biological components that enhance and stabilize the activity in the inhibition of Aβ aggregation as well. In our previous reports, we elucidated the inhibitory activity of various natural products against Aβ aggregation by estimating EC_50_ values in PBS [24,26,32]. In the future, it will become more important to evaluate the inhibitory activity in a solvent that contains a large number of biological components, such as culture medium.

We also successfully visualized the inhibitory effect of AxN on Aβ aggregation and deposition on SH-SY5Y human neuroblastoma cells using QDAβ nanoprobes (Figure 2). QDAβ was specifically used to visualize Aβ aggregation with high sensitivity. It also emits intense photostable signals, which allow the clear detection of Aβ aggregation over time, providing a precise and reliable method to monitor Aβ aggregation dynamics in live cells. Similarly, Aβ fibril formation on SH-SY5Y cells was monitored using the ThT assay (Figure 3). ThT is a specific dye that binds to the β-sheet of amyloid fibrils, increasing fluorescence intensity upon binding and assisting in the identification of fibril deposition [33,34]. Recent studies showed that AxN with 50 µM was able to inhibit 80% of Aβ fibril formation in rat pheochromocytoma (PC 12) cells [35] and reduced ThT fluorescence, indicating the inhibitory ability of Aβ deposition [6]. These findings suggest that AxN inhibits Aβ aggregation and deposition by impeding fibril formation. This confirms that AxN can attenuate the formation of Aβ fibrilization on neurons, which is a key feature of AD.

Several studies demonstrated the neuroprotective potential of AxN by mitigating Aβ-induced cytotoxicity in several cell lines. The pre-treatment with 50 µM of AxN reduced cytotoxicity induced by 1 µM Aβ [35]. Similarly, another study demonstrated that 100 µM AxN had an anti-cytotoxic effect on olfactory ensheathing cells (OECs) against 10 µM Aβ-induced cytotoxicity [36]. Other studies also emphasized the ability of AxN to attenuate critical cellular pathways and inhibit programmed cell death [37,38]. In the current study, AxN’s protective effect was assessed using apoptosis markers, which showed a strong protective effect against early apoptosis but not against late necrosis (Figure 4). Therefore, the results suggest that the neuroprotective effect of AxN may partially rescue programmed cell death rather than overall protection against cell death. The accumulation and fibrillation of Aβ promoted cell death and neuronal damage in SH-SY5Y cells. Neurotoxicity induced by Aβ aggregation influenced oxidative stress and mitochondrial dysfunction, leading to cell death and neuronal damage, which was inhibited by AxN in SH-SY5Y cells by regulating the apoptotic signaling pathway [39,40,41]. Previously, it was also shown that AxN notably reduced the percentage of apoptosis, but it could not reduce necrosis in PC12 and rat retinal ganglion cells (RGCs) [19,42]. This finding suggests that AxN may play a crucial role in mitigating neuronal damage and enhancing cell viability. The ability to suppress cell apoptosis is considered a way to prevent AD. A study reported that AxN is safe for human consumption without causing any harmful effects even at a high supplementation dose [15]. Considering these studies, it can be stated that AxN is a safe material that has a significant neuroprotective effect against cytotoxicity induced by Aβ fibril formation in several cell lines.

We also revealed that the cell motility inhibited by Aβ aggregation was reduced by AxN treatment (Figure 5 and Figure 6). The WHA is a collective assay that involves cell–cell interactions and cell migration in a two-dimensional confluent cell monolayer [43]. WHA was previously performed on a monolayer of human brain microvascular endothelial cells (hBMECs) to evaluate the inductive effect of Aβ deposition, showing that Aβ deposition inhibited cell speed and distance covered during migration [11,27]. Previous studies showed that AxN enhanced migration in human keratinocyte (HaCaTs) cells. It was also reported that AxN enhanced migration in the human keratinocyte cell line HaCaT, restored motility, improved effective cellular organization [44,45], and reduced migration in breast cancer cells, indicating its potential to limit metastasis [46]. These results suggest that AxN is not only an anti-inflammatory agent but also attenuates the reduced cell motility caused by Aβ aggregation in AD while acting as an anti-cancer compound.

Some limitations of this study need to be acknowledged. First, as we mentioned, SH-SY5Y human neuroblastoma cells were treated with 25 µM for 24 h as an AD cell model. This is the same Aβ concentration and aggregation time used previously to screen various inhibitors using the MSHTS system [24,25,26,32] and has the advantage of shortening screening times by promoting aggregation with a high Aβ concentration. However, the Aβ concentration of 25 µM is higher than physiological conditions, so this may not be a suitable AD cell model. A future challenge will be to explore experimental conditions that more closely mimic physiological conditions. In addition, although this study indicates an inhibition of the effect of AxN on necrosis and cell migration dynamics, there is a gap in understanding the exact mechanism, as findings are limited. Further comprehensive studies are required to address the role and mechanism of the neuroprotective effects of AxN.

Even though AxN showed promising results in cell models, it is challenging to study in vivo efficacy and its clinical application, as AxN has poor aqueous solubility and low oral bioavailability. To overcome these barriers, formulation strategies like nano-capsulation and lipid-based cyclodextrin complexes can be considered to improve solubility and absorption. However, it also requires limiting the formulation stability and safety concerns in the biological environment to ensure AxN’s clinical potential.

## 4. Materials and Methods

### 4.1. Materials

Human neuroblastoma SH-SY5Y cells were obtained from KAC (Kyoto, Japan). Dulbecco’s Modified Eagle’s Medium (DMEM), penicillin-streptomycin, dimethyl sulfoxide (DMSO) and 1,1,1,3,3,3-hexafluoro-2-propanal (HFIP) (Fujifilm, Wako, Osaka, Japan), AxN (010-2761, Fujifilm, Wako, Osaka, Japan), DMEM/F12, fetal bovine serum (FBS) and fibronectin (Gibco/Life Technologies, New York, NY, USA), human Aβ_42_ (4349-v; Peptide Institute Inc., Osaka, Japan), Hoechst 33342 (H1399, Invitrogen, Carlsbad, CA, USA), pSIVA-IABND (N,N′-dimethyl-N-(iodoacetyl)-N′-(7-nitrobenz-2-oxa-1,3-diazol-4-yl)ethylenediamine), and propidium iodide (PI) (APO004, Bio-Rad Laboratories, Inc., Tokyo, Japan) were purchased. Quantum dot—Aβ_40_ (QDAβ) was prepared according to our previous reports [13,47].

### 4.2. Cell Culture

SH-SY5Y cells were cultured in DMEM supplemented with 10% FBS and 1% penicillin-streptomycin. The confluent culture was washed with 1xPBS, the adherent cells were detached using trypsin at 37 °C, and centrifuged at 2000 rpm for 5 min at 37 °C. The required amount of cell solution was then dispensed to culture dishes and made up with DMEM solution.

### 4.3. Preparation of Aβ_42_

5 mg of Aβ_42_ was dissolved using 5 mL of HFIP in a vial. The vial was covered and left to stand for 1 h at room temperature, gently stirring every 10 min. The solution was sonicated at 25 °C for 10 min to monomerize the peptide. HFIP was allowed to evaporate on a clean bench. After evaporation, the peptide was dissolved in DMSO. The prepared 1 mM or 2.5 mM Aβ_42_ was dispensed into microtubes and stored at −80 °C. The samples were used immediately after thawing and were not subjected to refreezing.

### 4.4. MSHTS System

The MSHTS system was established according to our previous report [30]. Aβ was co-incubated with five different concentrations (final 0.32, 1.6, 8.0, 40, and 200 µM) of AxN in 2% DMSO and stained with QDAβ in two different MSHTS solvents (PBS and DMEM/F12), then incubated at 37 °C for 24 h in a 1536-well plate. The QDAβ-Aβ_42_ aggregates that formed in each well were observed by an inverted fluorescence microscope (TE2000, Nikon, Tokyo, Japan) equipped with a color CCD camera (DP72, Olympus, Tokyo, Japan) and an objective lens (Plan Fluor 4x/0.13 PhL DL, Nikon). SD values of fluorescence intensities of 40,000 pixels (200 × 200 pixels) around the central region of each well were measured by Image J 1.53e software (NIH, Bethesda, MD, USA). The EC_50_ graph was drawn using Prism 9 GraphPad LLC (San Diego, CA, USA) software.

### 4.5. Visualization of Aβ Aggregation in the Cell Line

A 96-well plate was coated with 1 mg/mL fibronectin and incubated overnight at 37 °C and 5% CO_2_. Using a micropipette, 10 × 10^4^ SH-SY5Y cells were then seeded in the 96-well plate and incubated overnight at 37 °C and 5% CO_2_. 25 nM QDAβ was dissolved in DMEM/F12 and mixed with 50 µM Aβ. AxN-Aβ solution was prepared to achieve concentrations (final 0.032, 0.16, 0.8, 4.0, and 20 µM of AxN with 25 µM of Aβ). The solutions were dispensed to respective wells containing cultured cells and incubated at 37 °C and 5% CO_2_ for 24 h. Aggregation was observed using a Ti-E inverted fluorescence microscope equipped with a color CMOS camera (DS-Ri2; Nikon) and an objective lens (20x, Nikon). The Mean Gray value of images was measured using ImageJ 1.53e software (NIH).

### 4.6. ThT Assay

SH-SY5Y cells were seeded and incubated in a 1 mg/mL fibronectin-coated 96-well plate for 24 h at 37 °C and 5% CO_2_. The confluent cell layer was treated with 25 µM Aβ alone and with 20 µM ThT-0.032–20 µM AxN and incubated 24 h at 37 °C, 5% CO_2._ The solutions were removed and fresh DMEM/F12 was dispensed. Readings were taken using a fluorescence microplate reader (SH-9000, Yamato, Tokyo, Japan) at 490 nm. Images were captured using a Ti-E inverted fluorescence microscope combined with NIS-Elements C 4.51 software (Nikon) and analyzed using ImageJ 1.53e software.

### 4.7. Apoptosis Assay

SH-SY5Y cells were seeded in the 1 mg/mL fibronectin-coated 96-well plate and incubated overnight at 37 °C and 5% CO_2_. AxN dilutions (0.032–20 µM in DMEM/F12) were prepared and mixed with Aβ containing pSIVA-IANBD (100x) and PI (200x), then incubated for 24 h at 37 °C and 5% CO_2_. Using a Ti-E inverted fluorescence microscope equipped with a DS-Ri2 and color CMOS camera, observations were made, and images were captured and analyzed using ImageJ 1.53e software.

### 4.8. WHA

A 96-well plate was coated with 1 mg/mL fibronectin, then incubated overnight at 37 °C and 5% CO_2_. SH-SY5Y cells were seeded in the 96-well plate using DMEM solution and incubated overnight at 37 °C and 5% CO_2_ until confluence was reached. The cell layer was scraped using a sharp tip to make a wound, and 0.032–20 µM AxN containing 25 µM Aβ was added. Using a Ti-E inverted fluorescence microscope equipped with a DS-Ri2 color CMOS camera, four randomly selected scraped areas were captured every 15 min during a 24 h observation period and analyzed using ImageJ 1.53e software.

### 4.9. Migratory Assay

A 96-well plate was coated with 1 mg/mL fibronectin, then incubated overnight at 37 °C and 5% CO_2_. SH-SY5Y cells were seeded in the wells and incubated at 37 °C and 5% CO_2_ for 24 h. SH-SY5Y cells were stained using 0.2% Hoechst and incubated for 1 h. 0.032–20 µM AxN dilutions with 25 µM Aβ were dispensed into respective wells. Using a Ti-E inverted fluorescence microscope equipped with a DS-Ri2 color CMOS camera (Nikon, Tokyo, Japan), four randomly selected scraped areas were captured every 15 min during a 24 h observation period. Images were analyzed using Fiji ImageJ 1.54f software (NIH). The directional persistence was measured by migration pass/total length, and cell speed was determined by total length/migration time. Graphs were plotted using Microsoft Excel software.

### 4.10. Statistical Analysis

The statistical data were analyzed using Microsoft Excel (version 2401). All values are expressed as the mean ± SD. Two-tailed independent Student’s *t*-tests were used to compare the two groups. A *p*-value of less than 0.05 was considered statistically significant.

## 5. Conclusions

Our study demonstrated that co-incubation of AxN with 25 µM Aβ was able to attenuate the effect of Aβ aggregation in SH-SY5Y cells, while also reducing the effect of Aβ-induced impairments in cell motility. Although AxN did not restore late necrosis, early apoptosis was significantly inhibited, demonstrating its potent neuroprotective ability. This study used human neuroblastoma SH-SY5Y cells as a cell model. Our results suggest that AxN could be a promising neuroprotective therapeutic agent in treating neurodegenerative diseases, such as AD. In the future, AxN may be one agent used to cure neurodegenerative diseases.

## Figures and Tables

**Figure 1 molecules-30-04271-f001:**
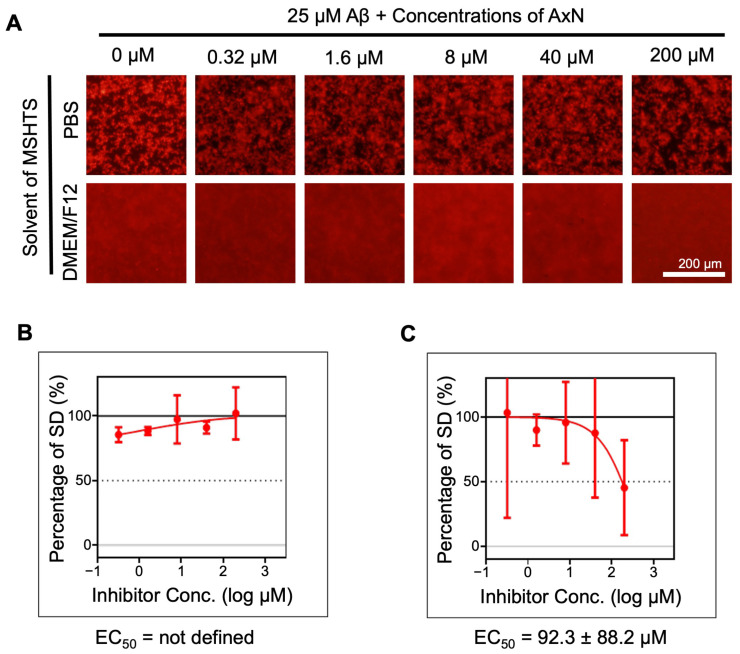
Inhibitory effect of astaxanthin (AxN) on Aβ aggregation. (**A**) Fluorescence images of 25 µM Aβ treated with different concentrations of AxN in two microliter-scale high-throughput screening (MSHTS) solvents. (**B**) Inhibitory activity of AxN on Aβ aggregation in PBS. (**C**) Inhibitory activity of AxN on Aβ aggregation in DMEM/F12. EC_50_ values were calculated from an inhibition curve. For each evaluation, values derived from images of four independent wells were used (*n* = 4).

**Figure 2 molecules-30-04271-f002:**
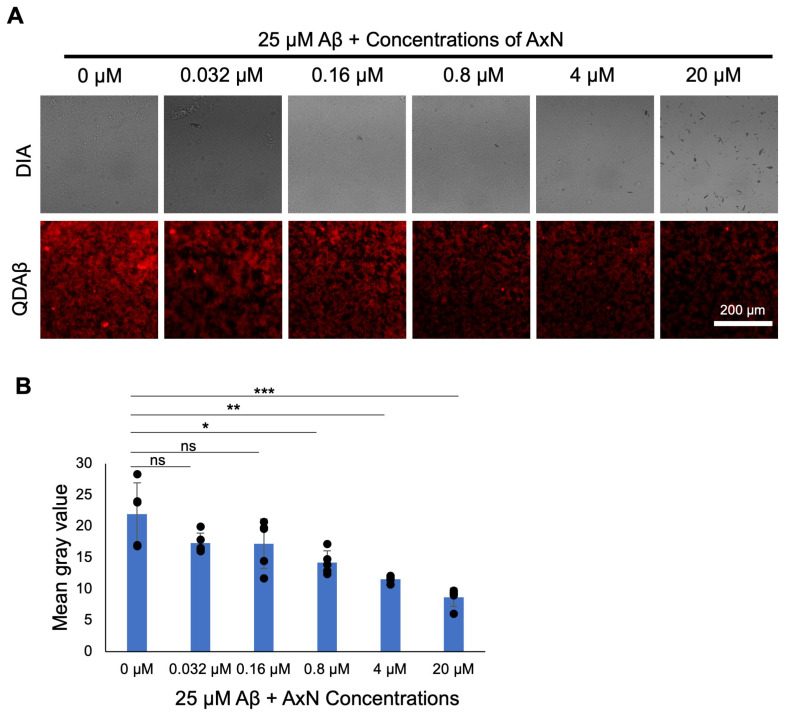
Attenuation of Aβ aggregation by astaxanthin (AxN) in SH-SY5Y neuroblastoma cells. (**A**) SH-SY5Y cells were co-incubated with 25 µM Aβ, 25 nM quantum dots—Aβ_40_ (QDAβ), and five concentrations of AxN, then were observed by an inverted fluorescence microscope. Digital image analysis (DIA) images are also shown in the figure. The dark particles seen in the high concentration of the AxN-treated sample are undissolved AxN. (**B**) Graphical representation of mean gray values of the control and each concentration of AxN. The data represent the mean ± SD from five independent fields of the same culture (*n* = 5). Statistically significant differences observed by the Student’s *t*-test (* *p* < 0.05, ** *p* < 0.005, *** *p* < 0.005, ns—not significant).

**Figure 3 molecules-30-04271-f003:**
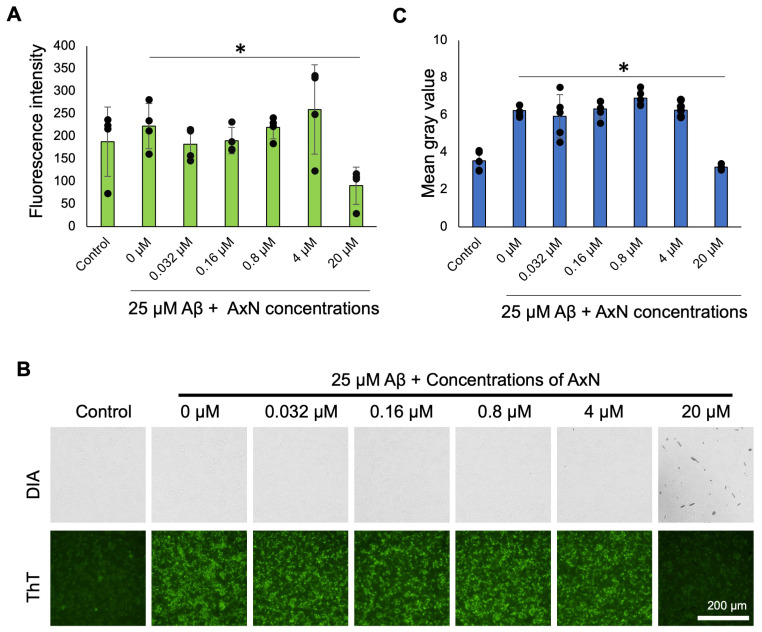
Inhibitory effects of astaxanthin (AxN) on Aβ deposition in SH-SY5Y neuroblastoma cells. (**A**) Fluorescence intensity of AxN concentrations at 24 h. The data represent the mean ± SD from four independent points of the same culture (*n* = 4). (**B**) SH-SY5Y cells were co-incubated with 25 µM Aβ, 20 µM Thioflavin T (ThT), and five concentrations of AxN and were observed by an inverted fluorescence microscope. Digital image analysis (DIA) images are also shown in the figure. (**C**) Graphical representation of mean gray value versus AxN concentrations. The data represent the mean ± SD from five independent points of the same culture (*n* = 5). Statistically significant differences observed by a Student’s *t*-test (* *p* < 0.05). To help readers’ understanding, the contrast of images in Figure 3B was adjusted using Microsoft PowerPoint software. The raw images are displayed in Appendix A.

**Figure 4 molecules-30-04271-f004:**
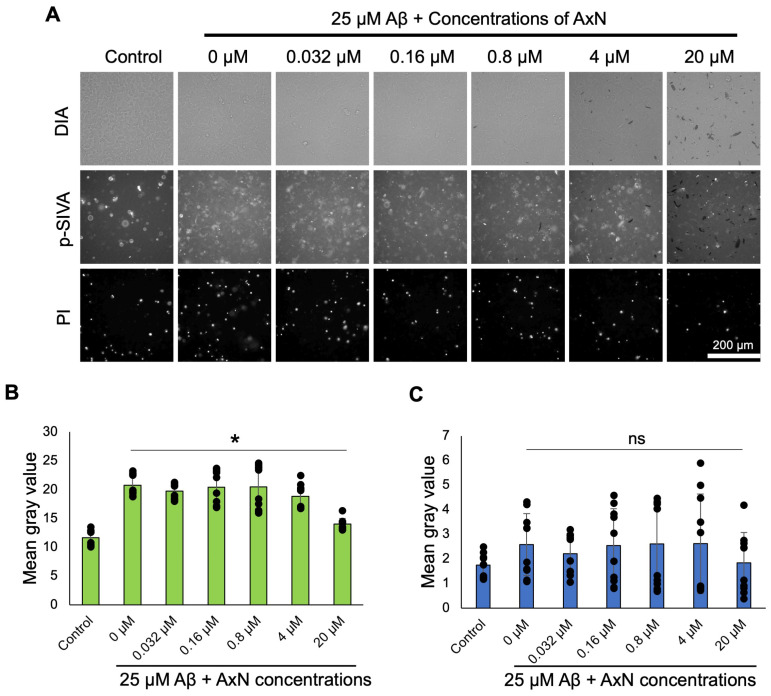
Effective prevention of early apoptosis but not late necrosis of SH-SY5Y neuroblastoma cells by astaxanthin (AxN). (**A**) Morphological apoptosis and necrosis were determined by staining with pSIVA-IAMBD and propidium iodide, respectively. Digital image analysis (DIA) images are also shown in the figure. The graphs represent early apoptosis (**B**) and late necrosis (**C**). The data represent the mean ± SD from five different fields of view of two independent experiments (*n* = 10). Statistically significant differences observed by the Student’s *t*-test (* *p* < 0.05, ns—not significant). To help readers’ understanding, the contrast of images in (**A**) was adjusted using Microsoft PowerPoint software. The raw images are displayed in Appendix A.

**Figure 5 molecules-30-04271-f005:**
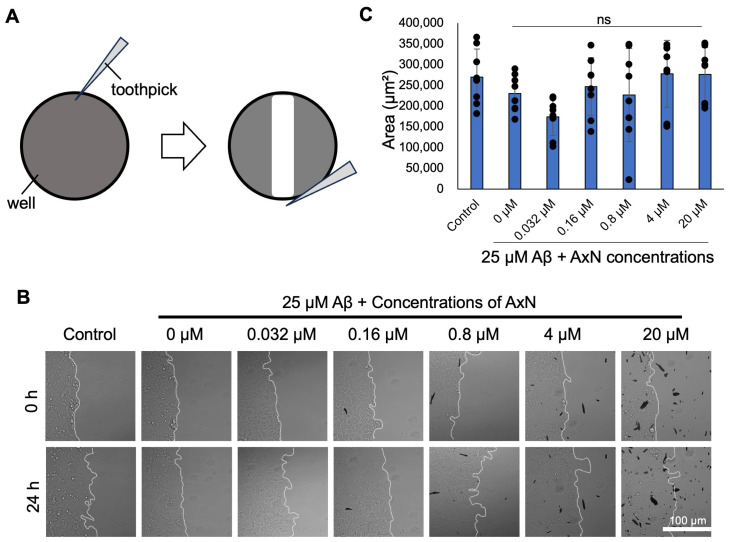
Effects of astaxanthin (AxN) on SH-SY5Y neuroblastoma cell motility. SH-SY5Y neuroblastoma cells seeded in a 96–well plate were treated with Aβ alone or with each concentration of AxN (0.032–20 µM) for 24 h. (**A**) Schematic representation of scratching the wound in a cell monolayer. (**B**) Images were captured using an inverted microscope, which demonstrates cell migration. (**C**) The area covered between 0 and 24 h for the control and each concentration of AxN. The data represent the mean ± SD from eight independent fields of two independent experiments (*n* = 8). Statistically significant differences observed by the Student’s *t*-test (ns—not significant). To help readers’ understanding, the scratches are highlighted, and the contrast of images in (**B**) was adjusted using Microsoft PowerPoint software. The raw images are displayed in Appendix A.

**Figure 6 molecules-30-04271-f006:**
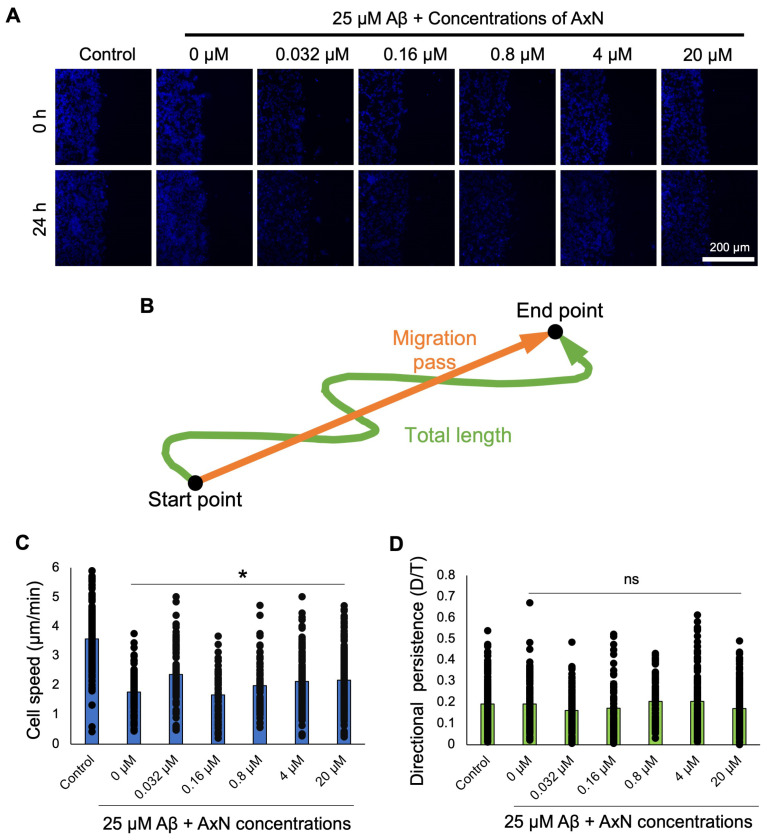
Effects of astaxanthin (AxN) on SH-SY5Y neuroblastoma cell migration using Hoechst staining. (**A**) Fluorescence images of 0.2% Hoechst-stained (for 1 h) SH-SY5Y neuroblastoma cells treated with Aβ alone or with each concentration of AxN (0.032–20 µM). (**B**) Schematic representation of cell migration from the initial to the end point. Graph demonstrates (**C**) the cell speed and (**D**) the directional persistence of cells. Cell movement was tracked for the following number of cells in each treatment group of a culture: control (*n* = 186), 0 µM (*n* = 100), 0.032 µM (*n* = 109), 0.16 µM (*n* = 66), 0.8 µM (*n* = 74), 4 µM (*n* = 149), and 20 µM (*n* = 194). Statistically significant differences observed by the Student’s *t*-test (* *p* < 0.05, ns—not significant).

## Data Availability

The original contributions presented in the study are included in the article/Appendix A, but further inquiries can be directed to the corresponding author.

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
