# Peer review of "Elucidation of the Neuroprotective Effects of Astaxanthin Against Amyloid β Toxicity in the SH-SY5Y Human Neuroblastoma Cell Line"

_molecules, 2025, doi:10.3390/molecules30214271_

Round 1
Reviewer 1 Report
Comments and Suggestions for Authors
The study presents new data regarding the capacity of astaxanthin (AxN) to reduce the neurotoxic effects associated with amyloid-β (Aβ) aggregation.
The authors employ a specific imaging approach based on conjugated quantum dots (QDAβ nanoprobes) to monitor Aβ aggregation in the presence or absence of astaxanthin (AxN), detecting a concentration-dependent decrease in aggregation. At this point, the authors also highlight the importance of physiological conditions to achieve the protective activity of AxN.
This quantum-dot imaging technique is also applied to SH-SY5Y neuroblastoma culture, where a decrease in Aβ aggregation is also observed. Also in this cellular model, the authors report that AxN treatment prevents early apoptosis, although it does not protect against late necrosis. Moreover, AxN enhances cell motility in cells treated with Aβ, supporting its protective effect.
Overall, the results of this work support the capacity of astaxanthin (AxN) to reduce the neurotoxic effects associated with Aβ aggregation. The study also suggests that this protective effect may be achieved by preventing early apoptosis and improving cell motility in neuroblastoma cells treated with Aβ.
Minor revision:
The manuscript is, in general, well written, and the text flows clearly, although several typographical errors should be corrected:
- In the title of Figure 4, the word “NO” is missing between “but” and “late necrosis.”
- In line 151, there is an extra point before (Figure 6A).
- In line 220, the sentence appears truncated around reference [15].
- In line 247, the term in vivo should be in italics.
The authors should proofread the manuscript to identify additional typographical errors and formatting inconsistencies. In particular, some references appear to be missing or incorrectly formatted, such as:
- A missing citation in line 143 (referring to the authors’ own previous work).
- An incorrectly formatted reference in line 278.
Major revision:
Regarding the figures, their overall quality could be improved. The panels within the same figure use different font styles and sizes. For example, in Figure 1, the font size in panel A is much larger than in panels B and C. In addition, the plots in panels C and E appear slightly blurry, and the axis fonts differ from those used in the rest of the figure. Figure 2 has a similar problem, where the font size in panel A is larger than in panel B, and the x-axis of panel B contains two different font types. In the following figures, these problems are not so evident, but consistency should be applied in font size, style, and resolution across all figures.
Regarding the statistical analysis described in the figures, the authors indicate the use of multiple Student’s t-tests and report the number of replicates. However, it is important to specify whether these represent independent biological replicates or, for example, individual fields from the same culture. In addition, since several concentrations are compared to a single control, a one-way ANOVA may be more appropriate to control against false positives due to multiple comparisons. Overall, the authors should clarify the level of replication (biological vs. technical) and justify the choice of the statistical test used.
Reviewer 2 Report
Comments and Suggestions for Authors
This interesting work addresses the potentially beneficial properties of AxN in inhibiting the formation of amyloid plaques. However, several issues remain to be clarified.
The authors utilize an in vitro model aimed at inhibiting the formation of Ab peptide aggregates. Such plaques form extracellularly during Alzheimer's disease (as opposed to tau protein-derived aggregates). In this study, however, the authors assess the formation of Ab aggregates both outside and inside SH-SY5Y cells. How, then, does the Ab peptide enter these cells? How do we know that the peptide was assessed inside the cells and the detected signal did not come from outside them? How does the method used differentiate this? Even if this issue has been addressed in other publications, it is worth including a few sentences of explanation in the text.
Please explain the role of quantum dots in this study? Why were they used? What was their function?
The authors should briefly describe how to interpret the obtained images (MSHTS method) presented in Figure 1A. Figure 1C indicates inhibitors of AxN activity on Aβ aggregation in DMEM/F12, but the images in Figure 1A do not support this in my opinion. How should they be interpreted?
Figure 2. Is AxN attenuating Aβ aggregation in SH-SY5Y neuroblastoma cells, or on (around) SH-SY5Y neuroblastoma cells? (See lines 87 and 89).
Line 131: How do the authors know that apoptosis (which was reduced by AxN) was caused by Ab aggregation, eg. in dose-dependent manner? Was an experiment conducted to demonstrate how varying degrees of Ab aggregation affect apoptosis in SH-SY5Y cells? Furthermore, the mere fact that a compound that is a potential drug candidate inhibits apoptosis in some cancer cells should be alarming and thoroughly commented on.
The scratch is not visible in the images in Figure 5B, or is very poorly visualized (?).
In lines 230-231, the authors, citing [47], state that "...AxN enhanced the migration of breast cancerous cells, restored 231 motility, and improved effective cellular organization..." while the cited publication shows the complete opposite (?). Please clarify, because if AxN increased cancer cell mobility, this compound would be dangerous in cancer patients.
Line 290: How much QDAβ was dissolved in DMEM/F12?
Round 2
Reviewer 1 Report
Comments and Suggestions for Authors
The authors have revised the manuscript and have addressed the issues raised in my previous review. I have no further comments, and I find the revised version suitable for publication.
Reviewer 2 Report
Comments and Suggestions for Authors
The authors answered all questions in the review. Their responses indicate that they are well-versed in the subject matter. The current version has been revised, and inaccuracies have been corrected. I have no further comments.